# MTSviewer: A database to visualize mitochondrial targeting sequences, cleavage sites, and mutations on protein structures

Andrew N. Bayne[1], Jing Dong[1], Saeid Amiri[2], Sali M. K. Farhan[2,3]*, Jean-François Trempe[1]*

**1** Department of Pharmacology & Therapeutics and Centre de Recherche en Biologie Structurale, McGill University, Montréal, Quebec, Canada, **2** Department of Neurology and Neurosurgery, Montreal Neurological Institute, McGill University, Montréal, Quebec, Canada, **3** Department of Human Genetics, McGill University, Montréal, Quebec, Canada

* jeanfrancois.trempe@mcgill.ca (JFT); sali.farhan@mcgill.ca (SMKF)

## Abstract

Mitochondrial dysfunction is implicated in a wide array of human diseases ranging from neurodegenerative disorders to cardiovascular defects. The coordinated localization and import of proteins into mitochondria are essential processes that ensure mitochondrial homeostasis. The localization and import of most mitochondrial proteins are driven by N-terminal mitochondrial targeting sequences (MTS's), which interact with import machinery and are removed by the mitochondrial processing peptidase (MPP). The recent discovery of internal MTS's—those which are distributed throughout a protein and act as import regulators or secondary MPP cleavage sites–has expanded the role of both MTS's and MPP beyond conventional N-terminal regulatory pathways. Still, the global mutational landscape of MTS's remains poorly characterized, both from genetic and structural perspectives. To this end, we have integrated a variety of tools into one harmonized R/Shiny database called MTSviewer (https://neurobioinfo.github.io/MTSvieweR/), which combines MTS predictions, cleavage sites, genetic variants, pathogenicity predictions, and N-terminomics data with structural visualization using AlphaFold models of human and yeast mitochondrial proteomes. Using MTSviewer, we profiled all MTS-containing proteins across human and yeast mitochondrial proteomes and provide multiple case studies to highlight the utility of this database.

## Introduction

Mitochondria are central to organismal health and regulate a diverse array of cellular processes, ranging from energy generation to immunity, proteostasis, and more [1–3]. Even though mitochondria contain their own genome, most mitochondrial proteins are nuclear encoded, translated in the cytosol, and imported into mitochondria [4]. Consequently, mitochondria have evolved an intricate system of targeting and translocation to import these proteins through translocases of the outer (TOM) and inner (TIM) mitochondrial membranes

**Data Availability Statement:** The MTSviewer database is freely accessible via https://neurobioinfo.github.io/MTSvieweR/. Source code is available at https://github.com/neurobioinfo/MTSvieweR, and

has been archived with Zenodo (DOI: 10.5281/zenodo.7768427). Contact: jeanfrancois.trempe@mcgill.ca.

**Funding:** A.N.B. is supported by a Canadian Institutes for Health Research (CIHR) Doctoral Fellowship. J.D is supported by a CIHR Canada Graduate Scholarship. This work was supported by a Canada Research Chair (Tier 2) in Structural Pharmacology to J.-F.T., as well as a Discovery Grant from the Natural Sciences and Engineering Research Council (NSERC) of Canada (RGPIN-2022-04042). The funders had no role in study design, data collection and analysis, decision to publish, or preparation of the manuscript.

**Competing interests:** The authors have declared that no competing interests exist.

and sort them into their correct subcompartment [5]. The most common targeting mechanism for matrix-localized proteins utilizes N-terminal mitochondrial targeting sequences (N-MTS), which form amphipathic helices and engage with TOM receptors before being passed through the TIM23 complex into the matrix [6]. In the matrix, N-MTS are cleaved off by the mitochondrial processing peptidase (MPP), which acts as a gatekeeper between import and overall mitochondrial quality control [7]. The breadth of import mechanisms expands considerably when considering proteins localized to the intermembrane space (IMS). These proteins typically lack an N-MTS and rely on disulfide trapping for localization [8]. Likewise, transmembrane (TM) proteins require a combination of accessory machinery and/or MTS's for their insertion and sorting [8]. It recently emerged that some imported proteins contain internal MTS's (iMTS), which bind to TOM70 to regulate import rates and may also contain secondary MPP cleavage sites [9, 10]. Furthermore, some proteins lacking an N-MTS still localize to and import into mitochondria via their iMTS's [11, 12]. The current model derived from experiments in yeast highlights TOM70 as the main receptor for these iMTS's, though there has not been any extensive profiling of iMTS's in humans to date. Briefly, TOM70 is composed of a transmembrane anchor at its N-terminus, followed by an N-terminal clamp domain that mediates its interactions with chaperones, such as Hsp90, and a C-terminal core domain that binds to mitochondrial preproteins [13]. Overall, TOM70 has emerged as a multifaceted OMM receptor that regulates mitochondrial protein import, safeguards against the proteotoxicity of aggregation-prone preproteins, and acts as a signaling hub to catalyze the innate immune response [14, 15]. Backes et al. were the first to characterize iMTS's in yeast mitochondrial proteins and proposed a "stepping stone" model in which imported precursor proteins bind to TOM70 via their iMTS's to adopt an import-competent, unfolded state and facilitate translocation [9]. As such, these iMTS's are critical for regulating both import rates and for preventing errant folding of domains which may block import, lead to cytosolic aggregation, or mislocalize partially folded domains at the OMM instead of their desired subcompartment. Overall, the landscape of iMTS-containing proteins remains poorly characterized in humans, and there are currently no databases to facilitate these studies.

Mitochondrial targeting and import are also innately linked to proteolysis, as mitochondria contain more than 40 proteases, coined "mitoproteases", which regulate proteostasis, MTS removal, stress responses, signaling, and more [16]. While MPP is the main protease implicated in N-MTS processing, other proteases act sequentially after MPP cleavage, including MIP, which removes an octapeptide, and XPNPEP3, which removes a single amino acid [17]. In specialized cases, other mitoproteases can regulate distal cleavages to drive signaling events, including PARL, a rhomboid protease which cleaves TM domains within the inner membrane [18, 19]. One example of a tandem MPP/PARL-cleaved protein is PINK1, a mitochondrial kinase that relies on its import and processing to either initiate or avoid the mitophagic cascade [20–22].

To facilitate the study of mitochondrial import and proteolysis, various tools have emerged, namely databases of mitochondrially localized proteins and prediction algorithms for sorting, MTS/iMTS propensity, and cleavage sites. Specifically, the most comprehensive mitochondrial database to date is MitoMiner v4.0, which contains gene-centric disease mutations, cleavage site predictions, and tissue expression for mitochondrial proteins across different species [23]. Databases such as HmtVar [24], MITOMAP [25], and MitoZoa [26] provide useful tools to investigate pathogenic mutations within proteins encoded by the mitochondrial genome, yet lack critical information on nuclear-encoded proteins whose cellular roles are dictated by their specific targeting to and cleavage within mitochondria. Other databases which focus primarily on compiling lists of nuclear-encoded proteins with evidence for mitochondrial localization include MitoP2 [27], MitoRes [28], and MitoProteome [29]. In terms of localization and

cleavage site predictors, various tools have emerged including iMLP [30], TargetP2.0 [31], MitoFates [32], TPpred3 [33], and DeepMito [34], which utilize orthogonal approaches to predict mitochondrial localization, presequence propensity, and/or cleavage sites [35]. Mass spectrometry experiments optimized for the labelling and enrichment of newly generated N-termini (neo-N-termini) have provided evidence for both canonical (*i.e.* MTS removal) and non-canonical (*i.e.* distal sites or N-terminal ragging) cleavage events within mitochondria [36–38]. These data can be submitted to TopFIND, an online database which serves as the gold standard for aggregating proteolytic evidence from *in vitro* experiments [39]. Overall, while the aforementioned tools are undoubtedly useful, they remain scattered across online web servers and software packages, and there is no harmonized resource to rapidly compare their results for a protein of interest.

From a structural perspective, recent work has revealed the structures of human TOM and TIM complexes [40, 41], and of an iMTS-TOM70 complex between human TOM70 and the SARS-CoV2 protein ORF9b [42, 43]. Still, how human MTS's engage with and are passed across the other translocase subunits remains unclear. The structure of human MPP in complex with MTS substrates also remains unknown, which makes it difficult to confidently predict the consequences of MTS variants on presequence processing. From a genetic perspective, comparing the phenotypes of non-synonymous mutations within MTS's, iMTS's, or near cleavage sites may provide key insight into both areas, yet there is no database for this kind of analysis. There are also currently no resources to rapidly compare the outputs of the numerous mitochondrial prediction algorithms, to compare these predictions with experimentally determined cleavage events, or to visualize MTS's within 3D protein structures. To this end, we hope to expedite the genetic and structural interrogation of human mitochondrial proteins and their MTS's with a novel database: MTSviewer (Fig 1). MTSviewer is a protein-centric interactive tool that allows users to map the features mentioned above on structures predicted by AlphaFold for both human and yeast mitochondrial proteins. Like MitoMiner, it connects human disease mutations and variants information with MTS and cleavage site predictions, but in addition provides iMTS propensities, experimentally determined cleavage sites, structural context, and enables users to upload their own variant lists. We demonstrate the usefulness of our tool by profiling the distribution of iMTS in their structural context and by describing 3 case studies.

## Materials and methods

### Database construction

The human mitochondrial proteome was downloaded from the MitoCarta 3.0 (1136 proteins) [44]. Additional annotations for the MitoCarta protein list were appended from the Integrated Mitochondrial Protein Index (Q4pre-2021) [23]. The yeast (*Sacchromyces cerevisiae*) mitochondrial proteome was derived from a high confidence dataset (901 proteins) [45]. Protein sequences were queried by UniProt ID and were submitted to: (1) iMLP–an internal MTS-like predictor using long short-term memory (LSTM) recurrent neural network architecture [30]; (2) TargetP2.0 –a presequence and cleavage site predictor using deep learning and bidirectional LSTM [31]. (3) MitoFates–a presequence and cleavage site predictor using support vector machine (SVM) classifiers [32]; (4) TPpred3 –a targeting and cleavage site predictor using Grammatical Restrained Hidden Conditional Random Fields [33]; (5) DeepMito–a sub-mitochondrial localization predictor using deep learning and convoluted neural networks [34]. For cleavage sites derived from N-terminomics, mass spectrometry data were aggregated from TopFIND 4.1 by Uniprot ID of both human and yeast proteins [39]. For variants and functional annotations of human proteins, dbNSFP v4.3a was parsed by Uniprot ID against

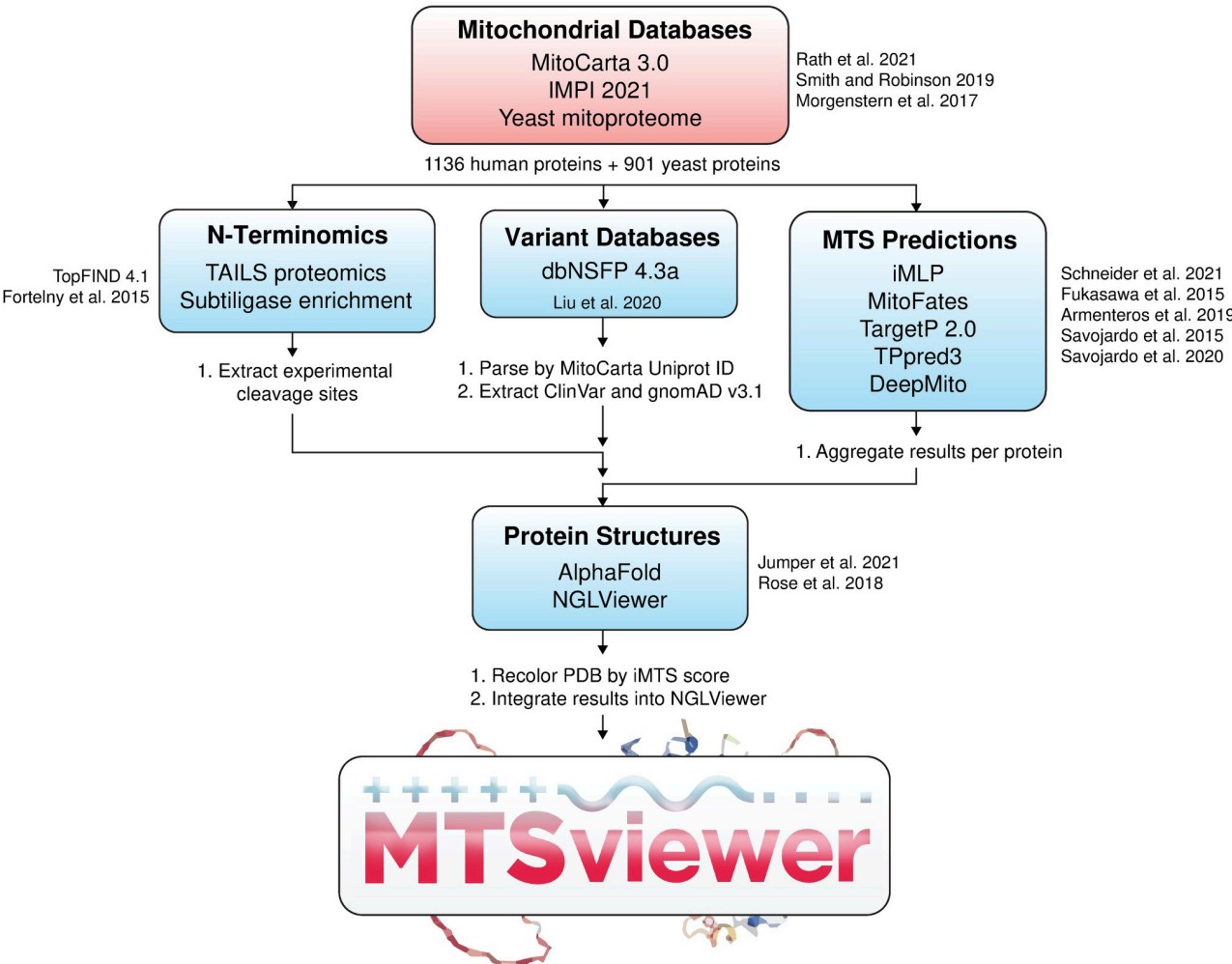

**Fig 1. Workflow of MTSviewer.** The database construction of MTSviewer, from initial mitochondrial databases to data integration and visualization.

GRCh38/hg38 coordinates [46]. The resulting list was filtered using an in-house Python script into separate datasets for gnomAD v3.1 and ClinVar. Variants unique to the ExAC database were ignored. AlphaFold models for the *Homo sapiens* proteome (UP000005640) and *Sacchromyces cerevisiae* (UP000002311) were downloaded and matched by Uniprot ID [47]. An in-house Python script based on BioPandas [48] was used to parse the PDB files and re-color B-factors according to iMLP scores (referred to interchangeably as iMTS scores) via iMLP. 3D visualization of protein structures was achieved using an adapted version of NGLViewer integrated into our R/Shiny application [49].

## Database utility

MTSviewer serves as a user-friendly platform for investigating MTS's from both genetic and structural perspectives. The database requires minimal bioinformatics knowledge and features both human and yeast mitochondrial proteomes. With MTSviewer, users are able to: (1) compare mitochondrial prediction outputs from a variety of algorithms; (2) visualize MTS likelihood on a folded protein structure; (3) compare experimentally identified and predicted proteolytic events; (4) map non-synonymous variants (gnomAD, ClinVar, or user uploaded)

within these MTS's and cleavage sites. Using this platform, we have also curated a list of disease-linked variants within human MTS's as a resource for their functional characterization.

## User interface

The MTSviewer user interface is intuitive and begins by selecting or searching a gene of interest. Users specify the desired database for variant visualization (currently gnomAD v3.1 or ClinVar), and variants are overlaid onto an XY plot with the iMTS probability from protein N- to C-terminus. Hovering over a variant reveals cursory details which are fully expanded in the variant table. For the structure viewer, two coloring schemes are toggleable: the iMTS score, or the AlphaFold per-residue predicted local distance difference test (pLDDT) confidence score. Users can investigate specific residues or variants by clicking on the iMTS plot or 3D structure, and the structure viewer will automatically highlight the interactions (*i.e.* polar contacts) and residues in proximity (5 Å) to the residue of interest (Fig 2). Users can also upload custom variant lists for their proteins of interest in CSV format, which will be added to the iMTS propensity curve, populated into the variant list data tables, and become visualizable on the 3D protein structure. This feature allows users to compare where their variants lie in terms of

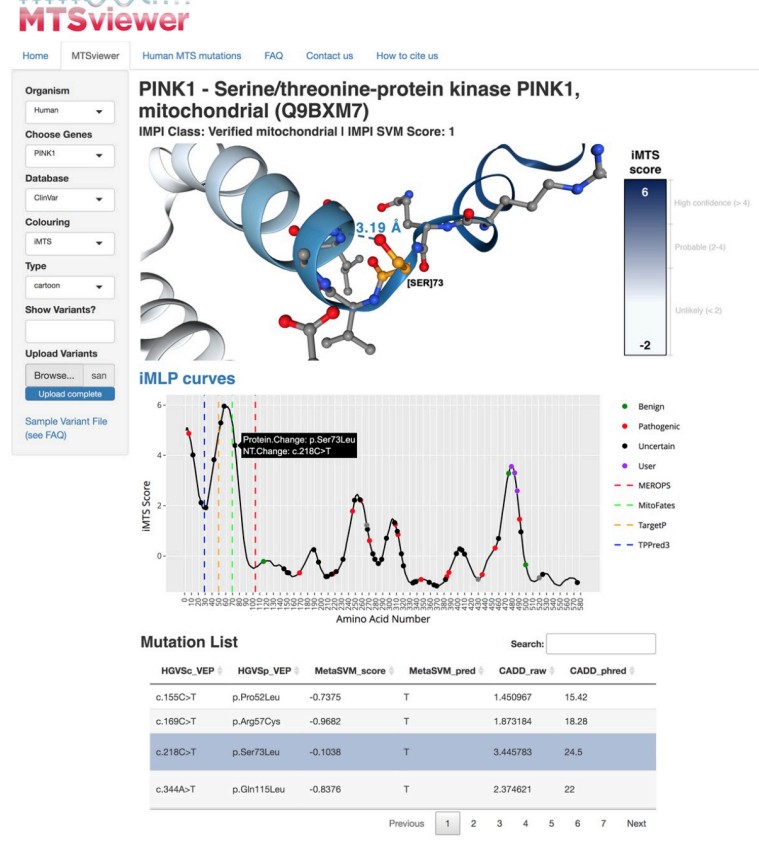

**Fig 2. MTSviewer output for PINK1.** A sample output from MTSviewer investigating human PINK1, a mitochondrial kinase with a uniquely long MTS and multiple predicted cleavage sites. Protein sequence, prediction algorithms, and N-terminomics data tables have been omitted for clarity but are available in full on the interactive MTSviewer web server. *Ser73Leu* has been highlighted as a variant of interest, as Ser73 is found within a region of high MTS propensity near the predicted MitoFates cleavage site.

MTS propensity, cleavage sites and other pathogenic variants on primary sequence and structural levels.

The iMTS plot and structure viewer also contain toggleable visualizations to highlight cleavage site predictions from the various MTS predictors and/or experimentally determined N-terminomics sites. Aggregated comparisons of targeting predictors are pooled in table format, and data frames are exportable to facilitate downstream analyses. Taken together, these features enable users to rapidly generate protein-level hypotheses to test *in vitro*, or to rationalize previous *in vitro* findings with import- or protease-specific context.

## Results and discussion

### PINK1 as a case study

To highlight the utility of MTSviewer we have chosen PINK1 as a case study (Fig 2), given its cryptic N-MTS and the innate coupling of its import and processing to gate its accumulation on the TOM complex. Briefly, PINK1 is known to be cleaved by the rhomboid protease PARL in the IMM at Ala103, which is validated by the N-terminomics outputs seen in MTSviewer. The precise MPP cleavage site within the PINK1 N-MTS remains unknown, though an MPP-cleaved PINK1 fragment accumulates upon PARL knockdown [50]. Based on the MTSviewer output for PINK1, there are many possibilities for the N-MTS MPP cleavage site, which will be critical to validate using *in vitro* assays, along with the effects of N-MTS variants (*e.g. Gly30Arg*, *Pro52Leu*, *Arg57Cys*, and *Ser73Leu*). While some of these PINK1 N-MTS variants are still cleaved by PARL in healthy mitochondria and accumulate following mitochondrial damage [51], their import rates and effects on MPP processing remain unstudied. Experiments which swap the PINK1 N-MTS with those from other mitochondrial proteins have shown that PINK1 can still be imported into mitochondria with chimeric N-MTS's, though PINK1 accumulation is prevented [52]. While many of these N-MTS PINK1 chimeras can still be imported, their specific rates of import have also yet to be measured. This suggests that f elements of mitochondrial proteins (and variants within these regions) will be critical to study beyond the context of binary import success or blockage.

Another useful feature of MTSviewer is the ability to gauge the length of a protein's N-MTS by looking at the iMTS propensity plots. For reference, it has been estimated that MTS's are usually 15–50 amino acids long [4], yet the PINK1 N-terminus exhibits high MTS propensity across its first 90 amino acids. As all of the MTSviewer iMTS data is available to download, users will be able to analyze global trends in MTS length and propensity across protein families to investigate the downstream consequences of longer or atypical N-MTS's within mitochondrial proteins. Beyond the N-MTS, the iMTS plot reveals a putative iMTS within the C-terminus of PINK1 (a.a. 460–500), which could regulate its import or processing rates at the mitochondrial surface. This is consistent with the knockdown of TOM70 reducing PINK1 import *in vitro* [53]. Furthermore, the PINK1 mRNA is co-transported with mitochondria [54], so it will also be important to investigate the role of TOM70 binding to PINK1 and this putative iMTS during translation and import. The MTSviewer output for PINK1 also highlights the need to consider the oligomerization status of proteins when investigating their monomeric AlphaFold structures. PINK1 is known to dimerize on the OMM following depolarization which could occlude its iMTS in the folded dimeric state [55, 56], even if partially unfolded PINK1 monomers bind to TOM70 upon import. Overall, MTSviewer will guide subsequent studies of atypically targeted proteins like PINK1 in the context of their MTS propensity, cleavage sites, and genetic variants.

## Global profiling of MTS's in human proteins

One caveat of the structure-based approach to MTSviewer is that N-terminal MTS's within AlphaFold predictions are typically unstructured and have low confidence (*i.e.* pLDDT) scores. This is congruent with the majority of N-MTS's being proteolytically removed prior to folding. Conversely, iMTS's beyond the N-terminus could exist in both high or low pLDDT regions, depending on the AlphaFold prediction. To visualize the global relationship between structured or folded regions of a protein and their corresponding iMTS propensity, we generated a density plot for pLDDT and iMLP values on a per-residue basis for the entire human mitochondrial proteome (Fig 3).

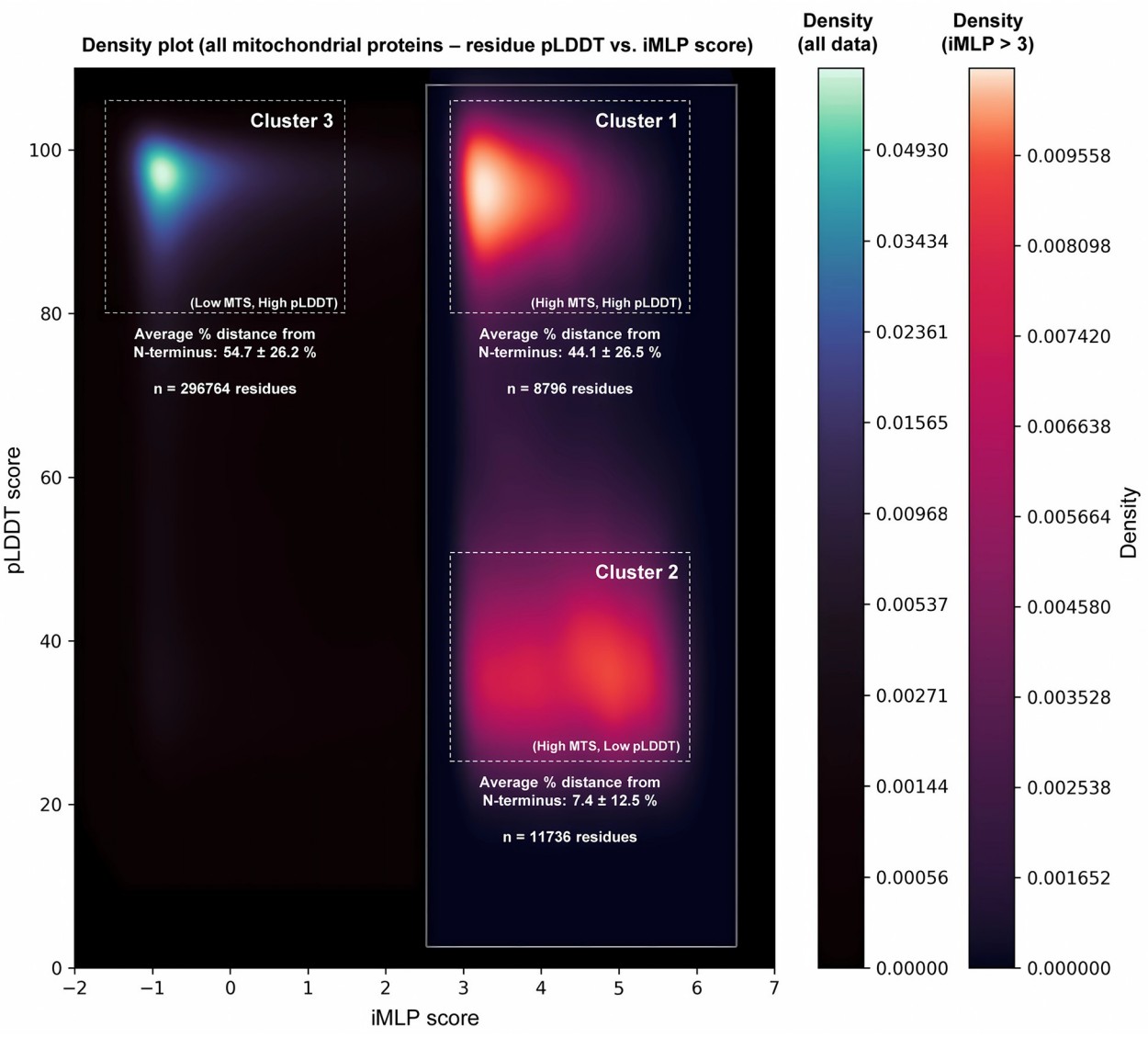

**Fig 3. Global analysis of per-residue iMLP and pLDDT scores across the human mitochondrial proteome.** AlphaFold pLDDT and iMLP scores were extracted on a per-residue basis (426832 residues total) for all 1136 proteins in the human mitochondrial proteome and were visualized as a kernel density estimate (KDE) plot using the seaborn library in Python. Two density plots were generated and merged to visualize low density datapoints—one for all pLDDT and iMLP scores (blue), and one for residues with high MTS propensity (iMLP score > 3) (red). Based on the KDE plot, data were grouped into three subsets (Cluster 1: pLDDT 80–100, iMLP > 3, Cluster 2: pLDDT 25–50, iMLP > 3, and Cluster 3: pLDDT 80–100, iMLP < 1.5) and the % distance of each residue within their corresponding protein length were calculated and written as the average % distance from N-terminus ± standard deviation for all residues in that cluster.

Of all residues, 69% were of high pLDDT values (pLDDT > 80) and low MTS propensity (iMLP score < 1.5) and were distributed within 25–75% of the protein sequence length (denoted cluster 3). This suggests that most residues in mitochondrial proteins are found in folded regions away from the extreme N- or C-termini and do not display significant MTS character. Next, we applied the same analysis to a subset of the data with iMLP scores > 3 in order to gauge what proportion of MTS's exist in folded or unstructured regions, and where these MTS's are found in the context of their protein sequence. From this subset of high MTS character residues, two distinct clusters emerged: one with high pLDDT scores (pLDDT > 80, denoted cluster 1), and one with low pLDDT scores (25 < pLDDT < 50, denoted cluster 2). Cluster 1 residues were found internally within protein sequences (average distance from N-terminus: 44.1%), while cluster 2 residues were found at protein N-termini (average distance from N-terminus: 7.4%). Taken together, this visualization provides quantitative evidence that (1) the majority of N-MTS's in an AlphaFold protein structure will be unstructured with low pLDDT scores; and (2) the majority of iMTS's within a protein will be found within a folded region of high pLDDT confidence.

## Structural analysis of iMTS-containing proteins

These conclusions above raise the question: why is it useful to visualize MTS's on a folded protein structure? To answer this, we first sought to characterize the iMTS's of all mitochondrial proteins (*i.e.* how many iMTS's does each protein contain and where are they located) to identify case studies. As such, we profiled the iMTS propensity curves of all mitochondrial proteins and counted their local maxima (defined as an amino acid with an iMLP score > 2 within a continuous 10 a.a. region of iMLP score > 1) using an in-house Python script (S1 File). To validate this approach, we first turned to the report that certain mitochondrial ribosomal proteins in yeast rely on iMTS's for their targeting and internalization into mitochondria [12]. Within our dataset, 18/54 (33%) of yeast mitochondrial ribosomal proteins (MRPs–chosen in our dataset by gene name containing MRPL or MRPS) contained two or more MTS's. In comparison, 34/78 (44%) of human MRPs contained two or more MTS's, which suggests that this atypical targeting is likely conserved in humans. In yeast, Backes et al. previously highlighted Atp1 (human gene *ATP5F1A*), the alpha subunit of the mitochondrial ATP synthase, as a model substrate for their "stepping stone" import via TOM70 [9]. While Atp1 contains an N-MTS, its mitochondrial import was shown to be dependent on the presence of 3 iMTS's–upon mutation of these iMTS's, both TOM70 binding and mitochondrial import were abrogated for Atp1. Our maxima counting dataset correctly identified the same 3 iMTS's of yeast Atp1, confirming the validity of our approach. In the AlphaFold structure of Atp1 on MTSviewer, these iMTS's are tightly folded and have high pLDDT scores, which supports the model that TOM70 must bind to and maintain these iMTS-containing precursors in a partially unfolded, import-competent state to prevent their premature misfolding and/or cytosolic aggregation. In humans, ATP5F1A contains two iMTS's, though specific import rates have not been studied. Other human proteins which contain multiple iMTS's include POLRMT (6 iMTS's), the mitochondrial RNA polymerase, and POLG (6 iMTS's), the mitochondrial DNA polymerase. Given that TOM70 has recently been shown to regulate the transcriptional activity of mitochondrial proteins to drive biogenesis [57], the import dependency of POLRMT on TOM70 will remain an important question to address. Other hits with multiple iMTS's include MAVS (3 iMTS's), the mitochondrial antiviral signaling protein, which has been shown to interact with TOM70 upon RNA viral infection to stimulate the innate immune response [58]. While the AlphaFold structure of MAVS is highly disordered, MTSviewer can be used as a platform to design *in*

*vitro* experiments that will clarify the role of these iMTS's in the formation of an antiviral signaling complex and/or localization of MAVS.

Next, we used our local maxima counting dataset to identify other proteins which contain C-terminal MTS's (C-MTS) by searching for iMLP maxima > 4 within the final 10% of their sequence (*i.e.* the extreme C-terminus). Within these hits was KMO (kynurenine 3-monooxygenase), a monooxygenase located primarily in microglia which regulates kynurenine metabolism and is implicated in neurological disorders including schizophrenia and Huntington's disease [59, 60]. Experimental studies confirmed the C-terminus dependent localization of KMO to the OMM in pig liver—specifically, mitochondrial localization of KMO was lost when truncated at a.a. 448, while its enzymatic activity was still retained [61]. From our maxima counting dataset, the C-MTS of human KMO is located between a.a. 420–460, with a maximum iMTS score of 5.2 at Tyr439. Using our structural visualization in MTSviewer, the KMO C-MTS is not predicted to be a part of the main KMO fold and as such could be solvent-accessible to bind TOM70 or anchor to the OMM even when the rest of the protein is folded, which corroborates the deletion experiments in pig KMO (Fig 4). KMO possesses a variant (*Arg452Cys*) which is located within this C-MTS and is associated with bipolar disorder, though the functional consequences of this mutation remain uncharacterized [62]. The

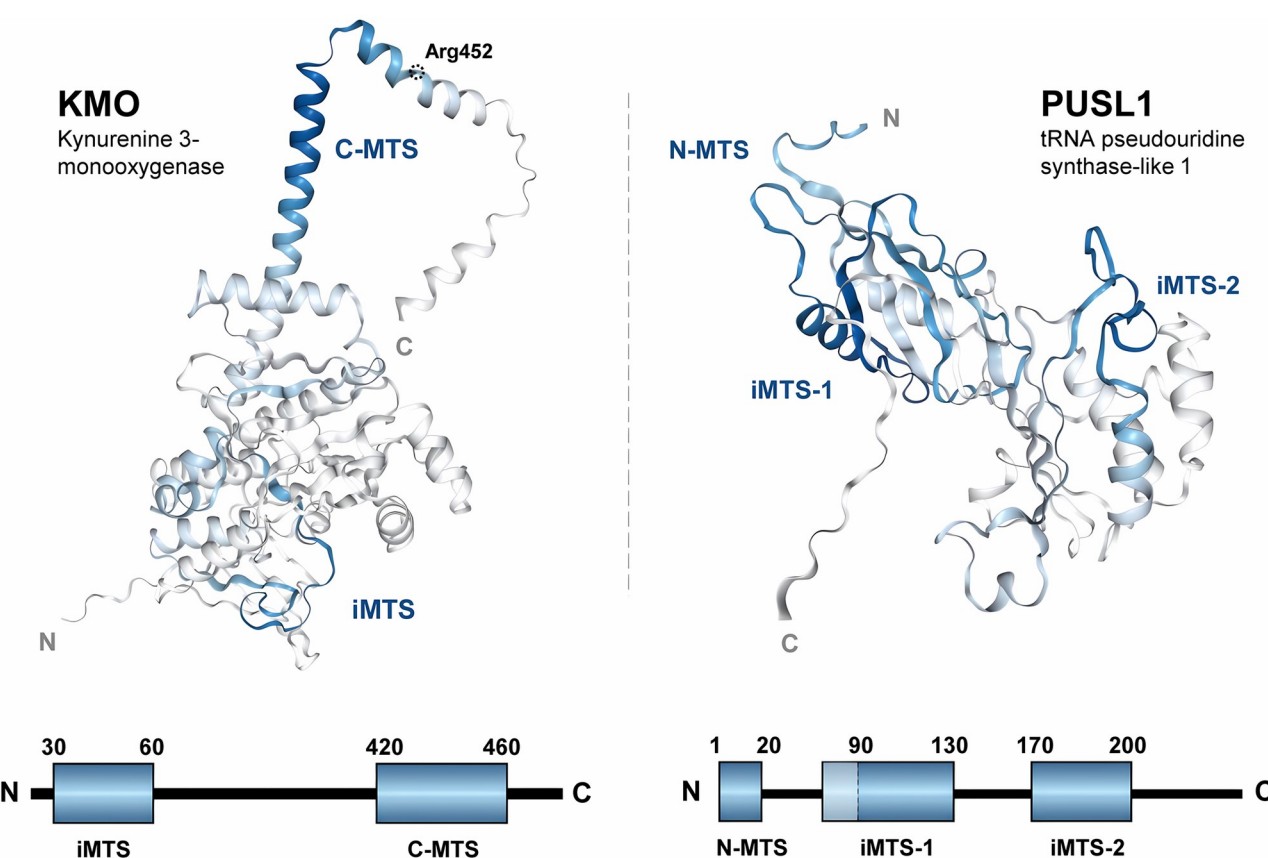

**Fig 4. Structural visualization of human KMO and PUSL1 iMTS propensity via MTSviewer.** AlphaFold models of KMO (Uniprot: O15229) and PUSL1 (Uniprot: Q8N0Z8) colored by iMLP score within MTSviewer. MTS's are annotated as blue rectangles on the sequence schematics. The C-MTS of KMO is predicted as helical, solvent-accessible, and is distinct from the folded domain of KMO. Arg452 is highlighted as a variant of interest (*Arg452Cys*). PUSL1 contains a putative N-MTS and two iMTS's (denoted iMTS-1 and iMTS-2), which are both found within the folded domain and would likely be inaccessible for TOM70 binding in this mature state.

dependence of KMO OMM localization on specific import receptors (*i.e.* TOM70) remains to be elucidated, but we hope that MTSviewer will facilitate these future explorations.

To find another example of a protein with high MTS propensity within a confidently folded region, we calculated the product of iMTS and pLDDT scores across the human mitochondrial proteome. The top hit from this calculation was PUSL1 (tRNA pseudouridine synthase-like 1), with its iMTS spanning a.a. 90–120 (average pLDDT = 97.7; average iMTS score: 5.0; local iMTS max at Arg103). PUSL1 is a recently characterized mitoribosome-interacting protein (MIP) in the mitochondrial matrix which associates with the IMM and regulates mitochondrial translation, most likely via its interaction with the 39S assembly intermediate [63]. Based on the structural visualization, the PUSL1 iMTS's would be inaccessible for TOM70 binding in its folded state, which suggests its potential reliance on the TOM70 "stepping stone" model for its import (Fig 4). Very little is known about the import regulation and structural function (*i.e.* which residues are required for its activity) of PUSL1, and as such PUSL1 remains an intriguing protein to study in the context of its targeting sequences and enzymatic function.

It is important to highlight that while regions of high iMTS propensity most likely bind to TOM70, high iMLP scores can also be suggestive of domains that bind to other chaperones or proteins with tetratricopeptide repeat (TPR) domains, as previously highlighted [30]. High iMLP scores also do not guarantee mitochondrial targeting–as shown in Bykov et al. 2022, the targeting of some yeast MRP's (MRPL15) required the presence of an N-MTS, while others (e.g. MRP20, MRP35) were sufficiently targeted by iMTS's in the absence of any N-MTS [12]. Also, in humans, creatine kinase B was shown to localize to mitochondria via its iMTS, though this localization was unchanged upon TOM70 silencing [11]. As such, *in vitro* experiments will remain critical to validate hypotheses generated with MTSviewer and to clarify the determinants for successful precursor targeting and TOM70 dependency in humans. With MTSviewer, users will be able to rapidly assess the distance between MTS's and folded regions, to validate the "stepping stone" model for proteins of interest *in vitro*, and to assess whether variants in these regions would affect TOM70-mediated import and/or function of the mature, folded protein. Overall, our platform will also facilitate the discovery and characterization of novel TOM70 substrates in humans.

## Conclusions

MTSviewer is a novel R/Shiny database for investigating the mutational space, targeting sequences, proteolysis, and 3D structures of mitochondrial proteins. Users require minimal bioinformatics training and can rapidly generate variant lists, investigate structural consequences, compare the results of various mitochondrial prediction tools, and dissect potential cleavage sites.

## Supporting information

**S1 File. Dataset of MTS counting across human and yeast mitochondrial proteomes.**
(XLSX)

## Acknowledgments

We would like to thank Niels van der Velden for the ongoing support with NGLVieweR scripting and integration.

## Author Contributions

**Conceptualization:** Andrew N. Bayne, Jean-François Trempe.

**Data curation:** Andrew N. Bayne, Jing Dong, Saeid Amiri, Sali M. K. Farhan, Jean-François Trempe.

**Formal analysis:** Andrew N. Bayne, Jing Dong, Saeid Amiri, Sali M. K. Farhan, Jean-François Trempe.

**Investigation:** Andrew N. Bayne, Jing Dong, Saeid Amiri.

**Software:** Andrew N. Bayne, Jing Dong, Saeid Amiri.

**Writing – original draft:** Andrew N. Bayne, Jean-François Trempe.

**Writing – review & editing:** Andrew N. Bayne, Jing Dong, Saeid Amiri, Sali M. K. Farhan, Jean-François Trempe.

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
