## [Decision Letter · Decision Letter 0]

31 Jan 2023

PONE-D-22-31504MTSviewer: a database to visualize mitochondrial targeting sequences, cleavage sites, and mutations on protein structuresPLOS ONE

Dear Dr. Trempe,

Thank you for submitting your manuscript to PLOS ONE. After careful consideration, we feel that it has merit but does not fully meet PLOS ONE’s publication criteria as it currently stands. Therefore, we invite you to submit a revised version of the manuscript that addresses the points raised during the review process.

We look forward to receiving your revised manuscript.

Kind regards,

Vivek Kumar, Ph. D.

Academic Editor

PLOS ONE

Journal Requirements:

Additional Editor Comments:

In this manuscript, the authors present MTSViewer, a database for the study of mitochondrial proteins that integrates a number of features such as cleavage predictions, pathogenicity predictions, N-terminomics, genetic variants, and structural visualization using AlphaFold models. They have integrated a number of tools and algorithms into an intuitive, user-friendly, online platform and the capabilities of the database including the workflow, user interface, and an example case study have been mostly described satisfactorily.

While the code and data are currently accessible from the public GitHub link shared by the authors, the contents on GitHub link can be revised or removed anytime. We would like to ensure that there is persistent access to them for PLOS ONE readers. Fortunately, there are numerous options available to facilitate this, which authors can exercise as per their convenience. The authors can preferably make the GitHub repository archivable using tools such as Zenodo. They can alternatively consider uploading the archived repository as a DOI artifact on a site like Mendeley and share the DOI in their revised submission. Or, they can also upload these files in a compressed/zipped file format as a supplement to this submission on PLOS ONE site. Eitherway, the authors should revise the submission with additional links based on the options discussed.

Also, an examination of the online repository suggests that some of the PDB files were omitted because of the directory size limit of 1000 files with the following message “Sorry, we had to truncate this directory to 1,000 files. 1,026 entries were omitted from the list.”

https://github.com/neurobioinfo/MTSvieweR/tree/main/MTSvieweR/PDBs/PDB_original

https://github.com/neurobioinfo/MTSvieweR/tree/main/MTSvieweR/PDBs/PDB_extracted

The authors can address this issue in a few different ways including archiving the PDB files into one or more zipped files to overcome the file count limit.

In addition, the authors are advised to address the valuable feedback below shared by the reviewer.

Bayne et al developed and present a novel database (MTSViewer) to consolidate predictions of MTS using multiple predictors, genetic variations, and predicted protein structures by AlphaFold. As described by authors, mitochondria play important roles in eukaryotic cells, and disturbance of targeting signal by genetic variants could potentially lead to dysfunction of this organelle. MTSViewer server is working well, and this platform could potentially be a useful resource for the community. I haven't seen any engineering or technical issues in the server. I feel however the authors should clarify some important aspects.

1. I understand that MTSViewer focuses on MTS (N-MTS and iMTS), however, it should be stated and argued in the context of mitochondrial protein database construction. The introduction section lacks referring to the other mitochondrial protein databases (only slightly mentioned in line 79), and it would be helpful if the currently existing databases are discussed in the section.

2. Comparison to similar databases section mainly discusses the novelty of MTSViewer, emphasizing integrating recent MTS predictors to connect variant information with such predictions. To my best knowledge, a similar approach was taken by another database MitoMiner (http://mitominer.mrc-mbu.cam.ac.uk/). Unfortunately, the database seems to be down for now, but MitoMiner collected genetic variants, pathogenicity information from OMIM, and MTS prediction results for mitochondrial genes. Indeed, MitoMiner was cited by the authors and incorporated into MTSViewer as described in line 105. The advantages and disadvantages of MTSViewer in comparison to MitoMiner (or another similar database) should be discussed as well rather than citing the database as one of the data sources.

3. The utilization of a predicted protein structure in MTSViewer is a bit unclear. As discussed in the main text, N-terminal MTS tends to have a low confidence score of AlphaFold and be shown as unstructured. Indeed, the region doesn’t have a high degree of conservation basically, inferring weaker evolutionary constraint for structuring than at least that for the main body. In contrast, it seems that showing the structure would be more important in the context of iMTS. Although the authors introduced Pink1 as a use case for N-terminal MTS and some iMTS, it would be helpful to discuss other interesting cases in the iMTS context and why structure information is important in this new database in addition to Pink1 case for N-MTS. I understand the C-terminal of Pink1 has a moderate iMLP score, however, it would convince more with an example having a higher score if it’s available.

Reviewers' comments:

Reviewer's Responses to Questions

**Comments to the Author**

1. Is the manuscript technically sound, and do the data support the conclusions?

Reviewer #1: Partly

2. Has the statistical analysis been performed appropriately and rigorously? 

Reviewer #1: N/A

3. Have the authors made all data underlying the findings in their manuscript fully available?

Reviewer #1: Yes

4. Is the manuscript presented in an intelligible fashion and written in standard English?

Reviewer #1: Yes

5. Review Comments to the Author

Reviewer #1: Bayne et al developed and present a novel database (MTSViewer) to consolidate predictions of MTS using multiple predictors, genetic variations, and predicted protein structures by AlphaFold. As described by authors, mitochondria play important roles in eukaryotic cells, and disturbance of targeting signal by genetic variants could potentially lead to dysfunction of this organelle. MTSViewer server is working well, and this platform could potentially be a useful resource for the community. I haven't seen any engineering or technical issues in the server. I feel however the authors should clarify some important aspects.

1. I understand that MTSViewer focuses on MTS (N-MTS and iMTS), however, it should be stated and argued in the context of mitochondrial protein database construction. The introduction section lacks referring to the other mitochondrial protein databases (only slightly mentioned in line 79), and it would be helpful if the currently existing databases are discussed in the section.

2. Comparison to similar databases section mainly discusses the novelty of MTSViewer, emphasizing integrating recent MTS predictors to connect variant information with such predictions. To my best knowledge, a similar approach was taken by another database MitoMiner (http://mitominer.mrc-mbu.cam.ac.uk/). Unfortunately, the database seems to be down for now, but MitoMiner collected genetic variants, pathogenicity information from OMIM, and MTS prediction results for mitochondrial genes. Indeed, MitoMiner was cited by the authors and incorporated into MTSViewer as described in line 105. The advantages and disadvantages of MTSViewer in comparison to MitoMiner (or another similar database) should be discussed as well rather than citing the database as one of the data sources.

3. The utilization of a predicted protein structure in MTSViewer is a bit unclear. As discussed in the main text, N-terminal MTS tends to have a low confidence score of AlphaFold and be shown as unstructured. Indeed, the region doesn’t have a high degree of conservation basically, inferring weaker evolutionary constraint for structuring than at least that for the main body. In contrast, it seems that showing the structure would be more important in the context of iMTS. Although the authors introduced Pink1 as a use case for N-terminal MTS and some iMTS, it would be helpful to discuss other interesting cases in the iMTS context and why structure information is important in this new database in addition to Pink1 case for N-MTS. I understand the C-terminal of Pink1 has a moderate iMLP score, however, it would convince more with an example having a higher score if it’s available.

6. PLOS authors have the option to publish the peer review history of their article (what does this mean?). If published, this will include your full peer review and any attached files.

Reviewer #1: No

---

## [Author Response · Author response to Decision Letter 0]

24 Mar 2023

Editor comments and author response 

In this manuscript, the authors present MTSViewer, a database for the study of mitochondrial proteins that integrates a number of features such as cleavage predictions, pathogenicity predictions, N-terminomics, genetic variants, and structural visualization using AlphaFold models. They have integrated a number of tools and algorithms into an intuitive, user-friendly, online platform and the capabilities of the database including the workflow, user interface, and an example case study have been mostly described satisfactorily.

While the code and data are currently accessible from the public GitHub link shared by the authors, the contents on GitHub link can be revised or removed anytime. We would like to ensure that there is persistent access to them for PLOS ONE readers. Fortunately, there are numerous options available to facilitate this, which authors can exercise as per their convenience. The authors can preferably make the GitHub repository archivable using tools such as Zenodo. They can alternatively consider uploading the archived repository as a DOI artifact on a site like Mendeley and share the DOI in their revised submission. Or, they can also upload these files in a compressed/zipped file format as a supplement to this submission on PLOS ONE site. Eitherway, the authors should revise the submission with additional links based on the options discussed.

We thank the editor for their comments and agree that persistent access to MTSviewer is essential. As suggested, we have archived the GitHub repository with Zenodo (DOI: 10.5281/zenodo.7768427). We have added the Zenodo DOI to the manuscript under the section “Availability of data and materials”. 

Also, an examination of the online repository suggests that some of the PDB files were omitted because of the directory size limit of 1000 files with the following message “Sorry, we had to truncate this directory to 1,000 files. 1,026 entries were omitted from the list.”

https://github.com/neurobioinfo/MTSvieweR/tree/main/MTSvieweR/PDBs/PDB_original

https://github.com/neurobioinfo/MTSvieweR/tree/main/MTSvieweR/PDBs/PDB_extracted

The authors can address this issue in a few different ways including archiving the PDB files into one or more zipped files to overcome the file count limit.

We have tested the offline download from GitHub and can confirm that all 2,000+ PDB files are downloaded correctly and work as intended. The directory size truncation is only a visual warning on the GitHub website and does not remove any of the files when the package is downloaded as a ZIP or pulled directly from GitHub. Still, we have uploaded an “AllPDBs.zip” in the PDB folder using GitHub Large File Storage.

Reviewer comments and author response 

Bayne et al developed and present a novel database (MTSViewer) to consolidate predictions of MTS using multiple predictors, genetic variations, and predicted protein structures by AlphaFold. As described by authors, mitochondria play important roles in eukaryotic cells, and disturbance of targeting signal by genetic variants could potentially lead to dysfunction of this organelle. MTSViewer server is working well, and this platform could potentially be a useful resource for the community. I haven't seen any engineering or technical issues in the server. I feel however the authors should clarify some important aspects.

1. I understand that MTSViewer focuses on MTS (N-MTS and iMTS), however, it should be stated and argued in the context of mitochondrial protein database construction. The introduction section lacks referring to the other mitochondrial protein databases (only slightly mentioned in line 79), and it would be helpful if the currently existing databases are discussed in the section.

We agree with the reviewer. We have added new examples of other mitochondrial databases for comparison and have merged the “Comparison to other databases” section (previously at the end of the text) directly into the introduction. These changes are now seen at lines 93-114.

2. Comparison to similar databases section mainly discusses the novelty of MTSViewer, emphasizing integrating recent MTS predictors to connect variant information with such predictions. To my best knowledge, a similar approach was taken by another database MitoMiner (http://mitominer.mrc-mbu.cam.ac.uk/). Unfortunately, the database seems to be down for now, but MitoMiner collected genetic variants, pathogenicity information from OMIM, and MTS prediction results for mitochondrial genes. Indeed, MitoMiner was cited by the authors and incorporated into MTSViewer as described in line 105. The advantages and disadvantages of MTSViewer in comparison to MitoMiner (or another similar database) should be discussed as well rather than citing the database as one of the data sources.

We thank the reviewer for this comment and agree it is critical to compare the utility of MTSviewer directly to MitoMiner. While the MitoMiner database is unfortunately still inaccessible, we used an archived version of the web page to access it. As in point #1, this direct comparison has also been included in the new introduction paragraph (commentary on MitoMiner can be found in lines 93-100 as well as 130-135). We also believe that our addition of new MTS profiling data and specific case studies (see question #3 below) implicitly highlights the utility of MTSviewer compared to MitoMiner and other pre-existing databases. 

3. The utilization of a predicted protein structure in MTSViewer is a bit unclear. As discussed in the main text, N-terminal MTS tends to have a low confidence score of AlphaFold and be shown as unstructured. Indeed, the region doesn’t have a high degree of conservation basically, inferring weaker evolutionary constraint for structuring than at least that for the main body. In contrast, it seems that showing the structure would be more important in the context of iMTS. Although the authors introduced Pink1 as a use case for N-terminal MTS and some iMTS, it would be helpful to discuss other interesting cases in the iMTS context and why structure information is important in this new database in addition to Pink1 case for N-MTS. I understand the C-terminal of Pink1 has a moderate iMLP score, however, it would convince more with an example having a higher score if it’s available.

We thank the reviewer for this comment and have adjusted the text substantially to address this. We added two new sections titled “Global profiling of MTS in human proteins” (line 245-282) and “Structural analysis of iMTS-containing proteins” (line 283-376). In this section:

A. We generated new data that directly shows the relationship between MTS score and AlphaFold confidence across the human mitoproteome (Fig 3), providing quantitative evidence to show that while N-MTS’s are mostly unstructured, there are also a significant amount of iMTS’s which exist in high-confidence regions of the AlphaFold predictions.

B. We created a new supplemental dataset that counts the number of MTS’s and their location in each protein (across both human and yeast mitoproteomes), both as a resource for the community and to identify new case studies to address this point. In the text, we use this dataset to highlight other human MTS-containing proteins (eg. POLRMT, MAVS, KMO, PUSL1) (line 307-376) and discuss the benefit of considering structural context when investigating MTS’s and mitochondrial import. 

C. We specifically highlight the structural context of KMO and PUSL1 MTS’s (Fig 4) and discuss the broader utility of MTSviewer, both in the context of previous work in yeast/TOM70 studies and as a resource for studying iMTS-containing proteins in humans. 

To accurately address all of these comments, we have added new references to the revised version of the manuscript: within the “Global profiling of MTS’s and other case studies” section (references #56-62) and within the introduction (references #13-15 and #24-29).

Overall, we believe that the reviewer highlighted key shortcomings in our article that we have sought to address with these new additions. We believe that we have substantially improved our manuscript thanks to the reviewer’s insightful commentary. Thank you for your consideration.

---

## [Editor Report · Decision Letter 1]

3 Apr 2023

MTSviewer: a database to visualize mitochondrial targeting sequences, cleavage sites, and mutations on protein structures

PONE-D-22-31504R1

Dear Dr. Trempe,

We’re pleased to inform you that your manuscript has been judged scientifically suitable for publication and will be formally accepted for publication once it meets all outstanding technical requirements.

Kind regards,

Vivek Kumar, Ph. D.

Academic Editor

PLOS ONE

Additional Editor Comments (optional):

The authors have made significant revisions in response to the the last round of review. Here are the key revisions:

They have included references to and comparison with other mitochondrial data sources and tools including MitoMiner. They have also included robust examples and use-cases to highlight the utilization of predicted protein structure in MTSViewer and have added new datasets, two figures and two sections in support of this.

They have uploaded additional PDB files to ensure that they are accessible. Also, they have persisted the raw data and supporting Python scripts used in the MTSviewer at GitHub using Zenodo.

The authors have also updated the organization and the format of the sections to improve the readability of the manuscript and have uploaded the figures as tiff instead of eps for improved cross-platform access.

The manuscript is being recommended for publication in the PLOS ONE. Congratulations!

---

## [Editor Report · Acceptance letter]

13 Apr 2023

PONE-D-22-31504R1 

MTSviewer: a database to visualize mitochondrial targeting sequences, cleavage sites, and mutations on protein structures 

Dear Dr. Trempe:

I'm pleased to inform you that your manuscript has been deemed suitable for publication in PLOS ONE. Congratulations! Your manuscript is now with our production department. 

Kind regards, 

on behalf of

Dr. Vivek Kumar 

Academic Editor

PLOS ONE